# Microcomputed Tomography in the Adriamycin Rat Model of Malformations—Preliminary Results

**DOI:** 10.3390/biom15040569

**Published:** 2025-04-11

**Authors:** Jan Scherberich, Leopold Bauer, Nana Huber-Liu, Gabriele Anja Krombach, Roman Metzger, Dietrich Kluth, Marco Ginzel

**Affiliations:** 1Department of Diagnostic and Interventional Radiology (Experimental Radiology), University Hospital Giessen, 35392 Giessen, Germany; jan.scherberich@radiol.med.uni-giessen.de (J.S.); gabriele.krombach@radiol.med.uni-giessen.de (G.A.K.); 2Department of Pediatric and Adolescent Surgery, Paracelsus Medical University Hospital, 5020 Salzburg, Austria; le.bauer@salk.at (L.B.); n.huber-liu@salk.at (N.H.-L.); r.metzger@salk.at (R.M.); 3Department of Pediatric Surgery, University Hospital Leipzig, 04103 Leipzig, Germany; dietrich.kluth@medizin.uni-leipzig.de

**Keywords:** Adriamycin, Doxorubicin, VACTERL, tracheal agenesis, cardiovascular malformations, microcomputed tomography, µCT, embryology, tracheoesophageal malformation

## Abstract

Background: Adriamycin, a chemotherapy drug, was first shown to induce malformations in rat embryos in 1978, with subsequent studies focusing on the development of esophageal atresia. Recently, we examined early tracheal development in healthy rat embryos using microcomputed tomography, which is useful for visualizing the morphology during embryonic development. Here, we use this technique to show malformations in rat embryos after Adriamycin treatment in the form of an atlas. Methods: Pregnant rats received Adriamycin [1.75 mg/kg body weight] intraperitoneally once daily from day 7 to 9. The embryos were harvested between the 16th and 21st embryonic day, fixed in Bouin’s solution and subsequently dried in a critical point dryer. The dried embryos were scanned using microcomputed tomography. Existing datasets of normal embryos served as controls. Results: The embryos affected by Adriamycin showed a broad spectrum of malformations that have only partially been described in previous publications. Our preliminary data show a large number of malformations that can be attributed to the VACTERL association (Vertebral, Anorectal, Cardiac, Tracheoesophageal, Renal, and Limb anomalies). In addition, cardiovascular malformations have been observed that may directly affect the esophageal/tracheal morphology. Conclusion: Micro CT imaging used in Adriamycin-affected embryos shows the full spectrum of induced malformations.

## 1. Introduction

In 1978, Daniel J. Thompson studied the toxic effect of Adriamycin (Doxorubicin) in developing rat embryos [1]. Adriamycin is a chemotherapy drug that belongs to the class of anthracycline antibiotics and is used as a cytostatic drug in the treatment of various types of cancer. He administered Adriamycin intraperitoneally to pregnant rats and examined different treatment intervals and doses for their effects on the induction of malformations. This model has since been widely used to study foregut anomalies, particularly esophageal atresia and tracheoesophageal fistulae, which resemble the human VATER or VACTERL association (Vertebral defects, Anorectal anomalies, Cardiac defects, Tracheoesophageal fistula/esophageal atresia, Renal abnormalities, and Limb anomalies) [2]. Adriamycin’s teratogenic effects are attributed to its interference with DNA replication and cellular proliferation during critical embryonic stages. Later, a number of studies used this model to analyze foregut malformations using histology and microdissection [3,4,5,6,7,8,9].

Recently, we have studied the dynamics of foregut development in normal rat embryos using microcomputed tomography [10]. Our results suggested that the respiratory tract is the result of the outgrowth of the lung primordium, supporting the model established by Zaw Tun in 1982 [11]. However, this observation could not explain the presence of the different forms of foregut malformations [12]. We concluded that an animal model was needed to study the development of these malformations. The Adriamycin rat model offers a tool to investigate such anomalies. Thus, we used this model to study the induced malformations using micro-CT to obtain a better insight into the morphology of abnormal embryos.

## 2. Materials and Methods

Ethics and animal housing: Animal care and experimental procedures were approved by the institutional review boards (state directorate Saxony, veterinary and food monitoring, applications: T14/15, T44/16, T13/18) (Federal Ministry of Education, Science and Research, application: 2022-0.466.985). This study involved neither wild animals nor samples collected from the field. Animals were housed at the preclinical research unit of the Paracelsus Medical University in Salzburg in rooms with a controlled temperature (22 °C), humidity (55%), and 12 h light–dark cycle. Food and water were supplied ad libitum.

Induction of malformations: Sprague Dawley rats (Janvier labs, Le Genest-Saint-Isle, France) were mated, and pregnancy was verified by the presence of a vaginal smear and weight gain during the first 7 days. Staging was performed by definition of the gestational age, with the day of positive vaginal smear defined as embryonic day 0 (ED 0). 1.75 mg/kg BW Adriamycin (Accord Healthcare B.V. Utrecht, The Netherlands) was administered intraperitoneally once daily from ED 7 to 9 [13].

Sample preparation: Pregnant rats were euthanized with pentobarbital (800 mg/kg body weight) overdose. Embryos were harvested, fixed in Bouin’s solution for 3 days at room temperature, and stored in 80% ethanol. Overall, 10 embryos, regardless of their sex, aged from ED16 to ED21 were analyzed for the current study. To enhance image contrast, complete embryos were dehydrated with the “critical point” drying technique utilizing the critical point dryer E3100 (Quantum Design AG, Marly, Fribourg, Switzerland) as described previously [14,15].

Micro-CT scanning: Each rat embryo was scanned using high-resolution micro-CT (SkyScan 1272, Bruker, Kontich, Belgium). Embryos were mounted on brass holders using Leit-C-Plast (Plano, Wetzlar, Germany). ED 16–17 samples were scanned with 55 kV and 90 μA and ED 18 and 21 samples were scanned with 40 kV and 200 µA without filter. All scans were performed with 6 averages and 0.2° increments in a 180° rotation. Images were reconstructed with the scanner software (NRecon 1.7.0.4; Bruker) using a modified Feldcamp algorithm [16] and converted to an 8 bitmap file format. Three datasets of normal untreated control animals were used from Publissio ZB MED Information Centre of Life Science at https://doi.org/10.4126/FRL01-006424446 [14].

Segmentation: Embryonic structures were manually segmented using CTAnalyzer (CTAn^®^, v1.16.1.0, Bruker) by defining regions of interest (ROIs) around the embryonic structure. After data segmentation, the 3D viewing software CTvox^®^ (Bruker) was used to produce volume rendering and virtual sections for visualization. The segmentation of the embryonic organs to create 3D reconstructions was performed by M.G. (who has 9 years of experience with micro-CT scans of embryos). The analysis of the findings from the micro-CT datasets of malformed embryos was performed by D.K. (who has 42 years of experience in pediatric surgery and 48 years of experience in embryology for pediatric surgeons) and M.G. Cardiac malformations were assessed by Dr. med. Werner Siekmeyer, a specialist in pediatric cardiology (Clinic and Polyclinic for Cardiology, Leipzig, Germany).

## 3. Results

In our preliminary study, we obtained a total of 31 embryos with gestational ages of 16, 17, 18, and 21 days from four Adriamycin-treated dams (Table 1). Out of these, ten embryos (three per group from ED 16–18, plus 1 malformed ED 21 embryo) were selected for micro-CT scanning and analysis.

For micro-CT scanning, three embryos were randomly selected from ED 16 to ED 18 from each dam. In addition, an ED 21 rat embryo that was visibly malformed from the outside was included. The resulting ten datasets were used to gain an initial insight into the induced malformations and to visualize them. All of these embryos showed malformations of various degrees, with an average of 7.6 malformations per embryo (Table 2). We compared normal and Adriamycin-treated embryos in each age group and documented the morphological changes. Various malformations according to the VACTERL association were found (Figure 1) and a summary of observed malformations per embryo is shown in Table 2.

In addition, malformations of the midgut (multiple atresia of the midgut, stomach atresia and missing stomach) and completely missing thymuses were consistently observed.

In this study, we limit ourselves to the findings on foregut and cardiovascular malformations.

### 3.1. Esophageal Atresia

Esophageal atresia is the congenital partial absence of the esophagus, often combined with a tracheoesophageal fistula (Type 3b after Vogt [17]). Of the embryos examined, only ED 18#2 and ED 21#1 had esophageal atresia. Both showed a blind sac in the area of the hypopharynx/proximal esophageal region and a tracheoesophageal fistula (Figure 1 and Figure 2). Additionally, we found the following:

-Anomalies of the heart and great vessels.-Duodenal stenosis.-Malformations of the kidneys (hydronephrosis).-Additional malformations of the gut (anorectal atresia/agenesis).

### 3.2. Tracheal Agenesis

Tracheal agenesis is the congenital absence or underdevelopment of the trachea, which normally connects the larynx to the bronchi. Abnormal development of the trachea can range from partial agenesis to complete absence of the trachea (Type I–III, as described by Floyd et al. [18]).

So far, all ED 16 (#1–3) and ED 17 (#1–3) specimens and two ED 18 (#1,2) specimens examined have shown forms of tracheal agenesis. The morphology is similar to type III tracheal agenesis (according to Floyd), but differs in that there is esophageal tissue in the upper part and tracheal tissue in the lower part (Figure 1 and Figure 2). An upper tracheal blind sac connected to the larynx has not been observed in any of the affected specimens examined.

In the case of the samples of ED 16#2 and ED 18#3, there was an esophageal fistula connecting an underdeveloped stomach to the right bronchus (Figure 2). The other examined Adriamycin-treated specimens showed a complete absence of stomach and lower esophagus; thus, a lower fistula was missing. The lungs of Adriamycin-treated embryos were underdeveloped and partially malformed, with abnormal or missing pulmonary fissures (Figure 3). Stenoses in the transition zone between esophageal and tracheal tissue were observed in ED 16#3 and ED17#1 samples (Figure 3).

### 3.3. Cardiovascular Anomalies

A variety of cardiovascular anomalies were observed in almost all samples. Compared to healthy embryos of the same age, all of which had a left-sided aorta/aortic arch, embryo ED 16#2 treated with Adriamycin showed a right-sided/descending aorta, while embryo ED 17#1 exhibited an abnormal aortic arch resembling an aneurysm (Figure 4). Two embryos (ED 16#3, ED17#2) showed the formation of an abnormal vascular ring, formed by the aortic arch and the pulmonary artery that is connected by the ductus arteriosus, thus surrounding and narrowing the tracheoesophagus (Figure 4). The ductus arteriosus, which connects the pulmonary artery to the aorta, also showed irregularities in Adriamycin-treated embryos, depending on the course of the aortic arch and the pulmonary artery. As a result, irregular origins of the brachiocephalic, left common carotid, and left subclavian artery from the ductus arteriosus can be observed. Most hearts appeared to have turned leftward (Figure 5), though we found internal cardiac anomalies difficult to assess.

## 4. Discussion

In this study, we used the Adriamycin rat model to induce various malformations consistent with the human phenotype of VACTERL association [2]. In previous studies, either histology was used for evaluation or macroscopic photographs or dissected embryos were shown [3,4,5,6,7,8,9]. Here, we examined Adriamycin-treated rat embryos using micro-CT imaging and provided a better view of the phenotype of the described malformations through 3D reconstructions. The morphological representations are ideal for compiling a 3D atlas of tracheoesophageal and associated malformations.

Recently, we have studied the development of the foregut in rat embryos using microcomputed tomography. Our results are consistent with Zaw Tun’s outgrowth model of respiratory tract development, as the observed forms of tracheal agenesis in this study cannot be explained by incorrect separation of the foregut. We saw that in all cases of tracheal agenesis, the larynx was always present. This is also true for all forms of human tracheal agenesis. However, a tracheal sac connected to the larynx was not observed in our specimens and has not been reported in humans either [12,18]. Thus, the tracheobronchial development seems to be independent of laryngeal development. This contradicts previous studies by Merei et al., who referred to parts of the developing larynx as the atretic proximal trachea or tracheal bud [5,6]. Our observations strongly indicate that the paired lung primordium is the starting point not only of the bronchi but also of the trachea itself. Therefore, the lung buds remain attached to the esophagus when tracheal development is disturbed. Delayed outgrowth of the trachea would result in tracheal agenesis, with the resulting esophagus representing proximal esophageal tissue and distal tracheal tissue (Figure 6). It is reasonable that the cytostatic drug Adriamycin may delay tracheal outgrowth in this model.

In two cases, we observed esophageal stenosis in embryos affected by tracheal agenesis. Both stenoses were accompanied by vascular rings. It is therefore reasonable to assume that the vascular rings could be the cause of the development of these stenoses. Interestingly, the transition between esophageal and tracheal tissue was located in the area of these stenoses. While the vascular malformations were relatively easy to detect, internal cardiac anomalies appear to be very difficult to assess using micro-CT imaging, although Happel et al. showed cardiac malformations in chicken embryos using this technique [19]. The reason for this difficulty could be that conventional assessment of cardiac malformations is performed with a four-chamber view in fetal echocardiography (Doppler echocardiography) [20]. In contrast, micro-CT imaging is static and lacks the dynamic component of echocardiography. However, our study could not be performed on living samples, and echocardiography may not be feasible for such small embryos. Contrasting embryos with Lugol’s solution, as described in several studies [21,22,23,24,25], did not appear to be superior to the drying method we used. An alternative method that could offer more detailed morphological insights into abnormal heart development is scanning electron microscopy (SEM). Brosig et al. found that while SEM is superior for revealing detailed surface morphology, micro-CT imaging excels in visualizing undissected structures beneath the surface when used on identical embryos [15]. However, utilizing SEM to study cardiac defects in embryos requires highly specialized expertise in micro-preparation of embryonic hearts [26,27]. A completely different approach to examining cardiac cavities and potentially altered morphology in malformed hearts involves filling the heart with a silicone polymer (Microfil), which can then be scanned using micro-CT [28,29]. It is important to note that the assessment of cardiac malformations in the Adriamycin rat model may be complicated, as the observed malformations may not occur in humans. A review of the literature revealed that this model has not previously been used to investigate cardiac anomalies in embryos.

This study is presented as a preliminary study. Ultimately, we want to morphologically represent the entire development of Adriamycin-treated rat embryos from ED 11 to 1 day before birth (ED 21). However, the amount of malformations induced in this model is extensive. It has been demonstrated that toxic substances have a sensitive period during which various organs are affected by malformations at different stages [30]. For instance, Thalidomide has been shown to have a sensitive period between days 35 and 49. During this time, specific abnormalities can occur on certain days. The absence of ears and deafness is most likely to occur between days 35 and 37. The absence of arms typically happens between days 39 and 41. Phocomelia, where only three fingers are present, is most likely to develop between days 43 and 44. Lastly, thumbs with three joints are most commonly observed between days 46 and 48 [30]. It would be interesting to find sensitive periods for the various malformations seen in the Adriamycin model as well, as this would allow targeted injections to produce specific malformations.

## 5. Conclusions

The Adriamycin rat model exhibits various malformations that can be visualized and morphologically analyzed using microcomputed tomography. The main advantage of this technique is the possibility to examine all induced malformations virtually and without dissection of the embryo, thus excluding artifacts caused by microdissection. However, to improve this model, it would be desirable to find sensitive periods for the various malformations seen in the Adriamycin model, as this would allow targeted in-jections to produce specific malformations.

## Figures and Tables

**Figure 1 biomolecules-15-00569-f001:**
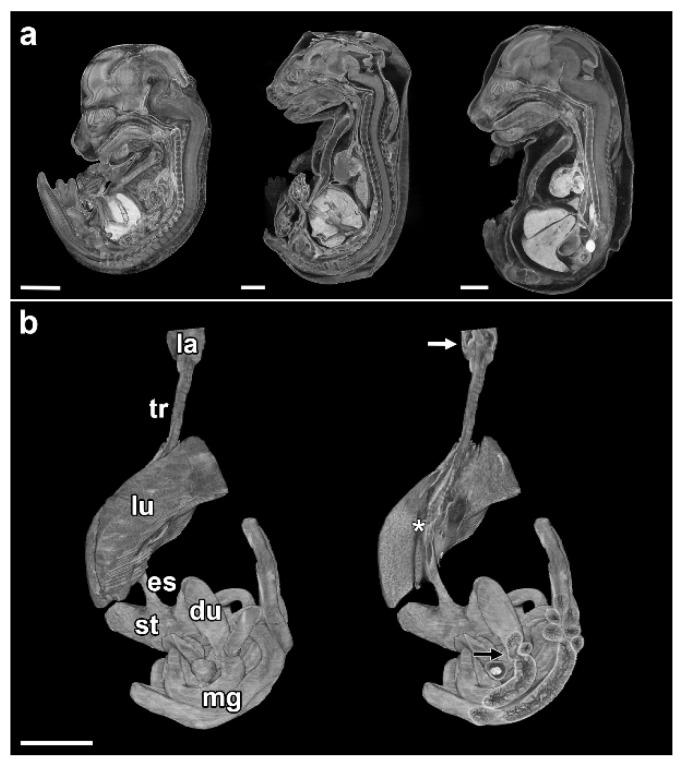
Overview of induced malformations in Adriamycin-treated rat embryos. (**a**) Lateral view on Adriamycin-treated rat embryos from embryonic day (ED) 16, 17, and 18 (from left to right). All embryos investigated had a missing thymus and bladder. (**b**) Virtually excised alimentary canal from the larynx downwards, including the lung of an Adriamycin-treated ED 21 rat embryo. The left shows the labeled reconstruction and the right shows partially broached segments, like the larynx, lung and duodenum, to show VACTERL-associated malformations. la: Larynx, tr: trachea, lu: lung, es: esophagus, st: Stomach, du: duodenum, mg: midgut, white arrow: esophageal blind sac, asterisk: tracheoesophageal fistula, black arrow: duodenal stenosis, scale bars = 2 mm.

**Figure 2 biomolecules-15-00569-f002:**
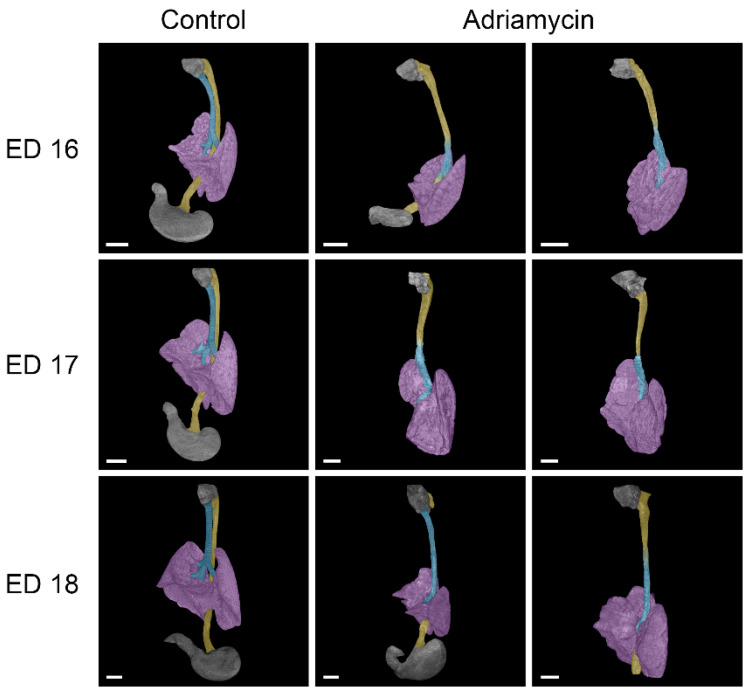
Lateral view of micro-CT reconstructions of the larynx, trachea, lungs, esophagus, and stomach in healthy and Adriamycin-treated rat embryos. Adriamycin treatment mainly led to tracheal agenesis type III, as described by Floyd [18], in our study. Notably, the connecting tube between hypopharynx and lung (purple) consisted of esophageal tissue (yellow) in the upper and tracheal tissue (turquois) in the lower part. One ED 18 embryo showed esophageal atresia with an esophageal blind sac at the dorsal larynx region (hypopharynx) and a tracheoesophageal fistula connecting the stomach to the right bronchus. However, most of the treated embryos had a missing stomach. Scale bars = 1 mm.

**Figure 3 biomolecules-15-00569-f003:**
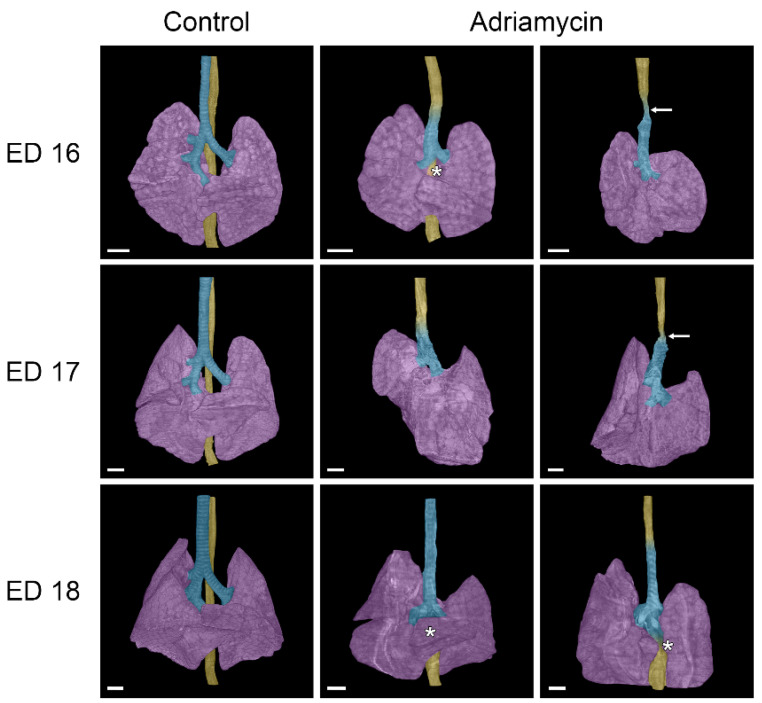
Front view of micro-CT reconstructions of the trachea, lungs, and esophagus in healthy and Adriamycin-treated rat embryos. Rat embryos treated with Adriamycin mainly show tracheal agenesis Type III, as described by Floyd [18], indicating a transition from the esophagus (yellow) into the trachea (turquois). In two embryos (ED 16 and ED 17), a stenosis of the tracheoesophagus is visible at this transition zone (arrows). A tracheoesophageal fistula (asterisk) to the right bronchus is visible in an ED 16 and two ED 18 Adriamycin-treated rat embryos. Purple: lung; scale bars = 500 µm.

**Figure 4 biomolecules-15-00569-f004:**
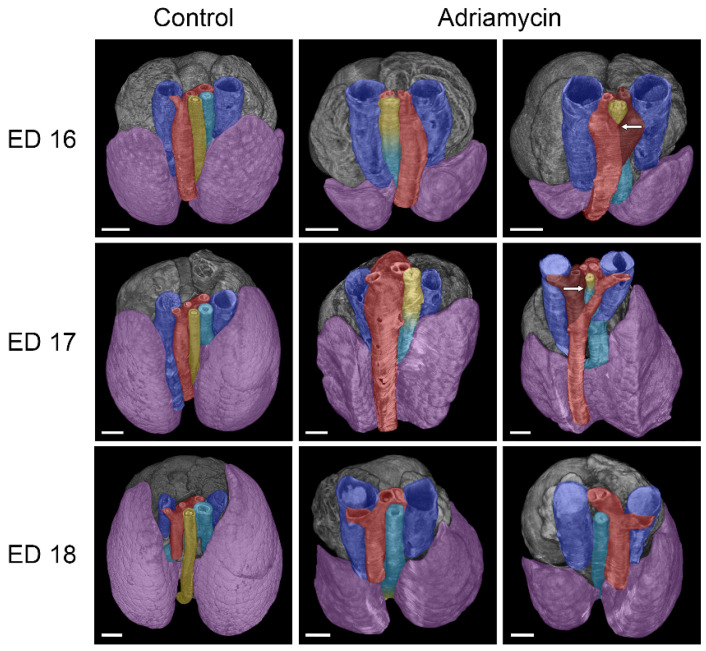
Dorsal view of micro-CT reconstructions of the heart, vessels, trachea, lungs, and esophagus in healthy and Adriamycin-treated rat embryos. Anomalies of the aortic arch were commonly observed in Adriamycin-treated rat embryos, ranging from a right-sided aortic arch, over “aneurysm-like” enlargements to a vascular ring surrounding the tracheoesophagus, consisting of the aorta (red) and the ductus arteriosus (dark red). The observed vascular rings at ED 16 and ED 17 are located in the area of the tracheoesophageal stenosis (arrows). Blue: left and right vena cava superior; turquoise: trachea; yellow: esophagus; purple: lung; scale bars = 500 µm.

**Figure 5 biomolecules-15-00569-f005:**
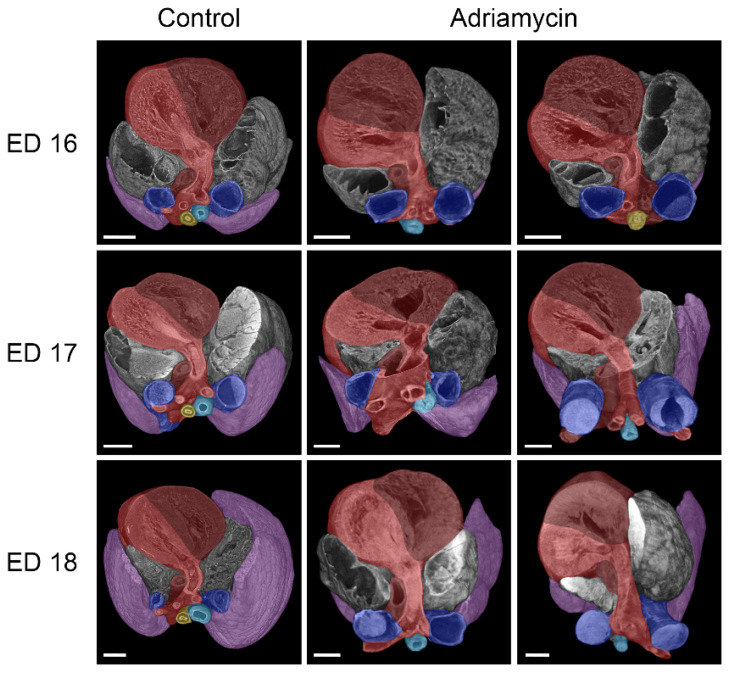
Cranial view of micro-CT reconstructions of the heart, vessels, trachea, lungs, and esophagus in healthy and Adriamycin-treated rat embryos. Most of the hearts of Adriamycin-treated rat embryos seem to be turned to the left compared to those of the controls. Red: aorta/left ventricle; dark red: pulmonary artery/right ventricle; blue: left and right vena cava superior; turquoise: Trachea; yellow: esophagus; purple: lung; scale bars = 500 µm.

**Figure 6 biomolecules-15-00569-f006:**
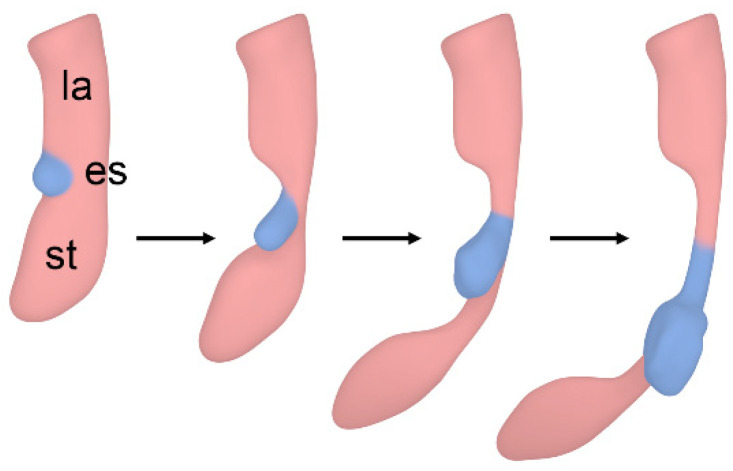
**Hypothetical course of events leading to tracheal agenesis.** Delayed outgrowth of the trachea can lead to tracheal agenesis, where the lung buds become attached to the developing esophagus. La: larynx; es: esophagus; st: stomach; blue: lung bud/developing trachea and bronchi.

**Table 1 biomolecules-15-00569-t001:** Distribution of embryos harvested (and used for this study) per gestational age group.

Gestational Age:	Embryos Harvested (Used):	Embryos Resorbed:
ED 16	12 (3)	1
ED 17	3 (3)	6
ED 18	6 (3)	4
ED 21	10 (1)	2

**Table 2 biomolecules-15-00569-t002:** Findings of various malformations in each investigated specimen. -: Not observed, X: observed, duod.: duodenal, sten.: stenosis, cardiov.: cardiovascular, mal.: malformation.

	ED 16			ED 17			ED 18			ED 21
	#1	#2	#3	#1	#2	#3	#1	#2	#3	#1
Esophageal atresia	-	-	-	-	-	-	-	X	-	X
Tracheal agenesis	X	X	X	X	X	X	X	-	X	-
Duod. sten./atresia	-	-	-	-	-	-	-	X	-	X
Missing bladder	X	X	X	X	X	X	X	X	X	X
Missing thymus	X	X	X	X	X	X	X	X	X	X
Missing stomach	X	-	X	X	X	-	-	-	X	-
Stomach atresia	-	X	-	-	-	X	X	-	-	-
Midgut atresia	X	X	X	X	X	X	-	-	X	-
Anal atresia	-	X	-	X	X	X	-	X	-	X
Cardiov. mal.	X	X	X	X	X	X	X	X	X	X
Kidney mal.	-	-	-	X	-	-	-	-	-	X
Lung mal.	X	X	X	X	X	X	X	X	X	X
Limb mal.	-	-	-	-	-	-	-	-	-	X
Total	7	8	7	9	8	8	6	7	7	9

## Data Availability

The data presented in this study are available on request from the corresponding author due to ongoing investigations and data collection. Once all datasets have been collected, they will be published together.

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
