# Peer review of "Microcomputed Tomography in the Adriamycin Rat Model of Malformations—Preliminary Results"

_biomolecules, 2025, doi:10.3390/biom15040569_

Round 1
Reviewer 1 Report
Comments and Suggestions for Authors
This experimental study shows evidences about abnormalities and malformations of rat embryos after in utero exposure to Adryamicin (Doxorubicin), at various gestational age. Three-dimensional reconstructions after micro-CT scan segmentations of high-resolution datasets provided new insights of already reported abnormalities using similar rat models, but only assessed by conventional (2D) histology.
I found this article original, well written, and definitely worth to be published on Biomolecules. Regarding sample preparation, I agree on the use of critical point dryer as a technique to improve image contrast as compared to fresh sample, even though several investigators prefer to stain their sample in Lugol solution for few days. I don't really get the reason of using different scan settings for ED 16/17 (55 kV, 90 uA) and for ED 18/21 (40 kV, 200 uA), but due to the qualitative nature of the subsequent image analysis, I exclude the presence of bias due to different scan settings between these groups. I only recommend authors to add a sentence in the Methods section about the training of the authors that have generated data in Table 2 (e.g., "image analysis was performed by X.Y., who is a pathologist with ZZ. years of experience", or similar).
Besides this minor amendments, my opinion is totally positive about this paper.
Author Response
We thank the reviewer for his/her review and would like to answer the questions and comments point by point.
Comment 1:
I don't really get the reason of using different scan settings for ED 16/17 (55 kV, 90 uA) and for ED 18/21 (40 kV, 200 uA), but due to the qualitative nature of the subsequent image analysis, I exclude the presence of bias due to different scan settings between these groups.
Response 1:
We thank the reviewer for his/her comment. Regarding the remarks about the different scan setting we provide the following explanation:
The different scan settings are the result of a technical adjustment to our specimens due to the use of a different micro-CT. The micro-CT scans of the controls were performed in a predecessor model of the currently used micro-CT. The successor model was not able to adopt the parameters initially used. Therefore, some adjustments of the parameters were necessary to achieve comparable results with the controls. In this study, all data sets with sufficient scan quality of the embryos were used, which explains the different scan parameters.
Comment 2:
I only recommend authors to add a sentence in the Methods section about the training of the authors that have generated data in Table 2 (e.g., "image analysis was performed by X.Y., who is a pathologist with ZZ. years of experience", or similar).
Response 2:
We thank the reviewer for this recommendation and we added the following statement in the method section (line 87-93):
The segmentation of the embryonic organs to create 3D reconstructions was performed by M.G. (9 years of experience with micro-CT scans of embryos). The analysis of the findings from the micro-CT data sets of malformed embryos was performed by D.K. (42 years of experience in pediatric surgery and 48 years in embryology for pediatric surgeons) and M.G. Cardiac malformations were assessed by Dr. med. Werner Siekmeyer, specialist in pediatric cardiology (Clinic and Polyclinic for Cardiology).
Reviewer 2 Report
Comments and Suggestions for Authors
In this manuscript, authors use microCT to show malformations in rat embryos after Adriamycin treatment in form of an atlas. The research provides insights into the morphological changes associated with Adriamycin - induced malformations, which may contribute to a better understanding of human congenital anomalies. However, before the manuscript could be published, some revisions should be made:
- Although the authors compare the findings to some previous studies, the comparison is not in - depth enough. A more comprehensive review of the literature would help to place the results in a broader context.
- The authors mention the difficulty in assessing internal cardiac anomalies using micro - CT. It would be beneficial to discuss potential solutions or alternative techniques that could be used to overcome this limitation. Additionally, the quality of the micro - CT images and the accuracy of the segmentation process could be further evaluated.
- Please include information on the power analysis used to determine the sample size, if any. For the segmentation process, mention any quality control measures taken to ensure accurate and consistent results.
Author Response
We thank the reviewer for his/her review and would like to answer the questions and comments point by point.
Comment 1:
Although the authors compare the findings to some previous studies, the comparison is not in - depth enough. A more comprehensive review of the literature would help to place the results in a broader context.
Response 1:
We thank the reviewer for this advice. We have re-reviewed the literature on esophageal and tracheal malformations in the embryonic rat model of Adriamycin and found no descriptions/ discussions comparable to our observations. In addition, we only show preliminary results, therefore an in depth discussion is currently difficult.
Comment 2:
The authors mention the difficulty in assessing internal cardiac anomalies using micro - CT. It would be beneficial to discuss potential solutions or alternative techniques that could be used to overcome this limitation. Additionally, the quality of the micro - CT images and the accuracy of the segmentation process could be further evaluated.
Response 2:
We thank the reviewer for this comment. We added the following paragraph (line 228-249):
While the vascular malformations were relatively easy to detect, internal cardiac anomalies appear to be very difficult to assess using micro-CT imaging, although Happel et al. showed cardiac malformations in chicken embryos using this technique [19]. The reason for this difficulty could be that the conventional assessment of cardiac malformations is performed by four-chamber view in fetal echocardiography (Doppler echocardiography) [20]. In contrast, micro-CT imaging is static and lacks the dynamic component of echocardiography. However, our study could not be performed on living samples, and echocardiography may not be feasible for such small embryos. Contrasting embryos with Lugol's solution, as described in several studies [21-25], did not appear to be superior to the drying method we used. An alternative method that could offer more detailed morphological insights into abnormal heart development is scanning electron microscopy (SEM). Brosig et al. found that while SEM is superior for revealing detailed surface morphology, micro-CT imaging excels in visualizing undissected structures beneath the surface when used on identical embryos [15]. However, utilizing SEM to study cardiac defects in embryos requires highly specialized expertise in micro-preparation of embryonic hearts [26,27]. A completely different approach to examining cardiac cavities and potentially altered morphology in malformed hearts involves filling the heart with a silicone polymer (Microfil), which can then be scanned using micro-CT [28,29]. It’s important to note that the assessment of cardiac malformations in the Adriamycin rat model may be complicated, as the observed malformations may not occur in humans. A review of the literature revealed that this model has not been previously used to investigate cardiac anomalies in embryos.
Comment 3:
Please include information on the power analysis used to determine the sample size, if any.
Response 3:
We thank the author for this reference. However, the purpose of this study was not to show the percentage of malformations caused by Adrimycin, but to generate malformed rat embryos to investigate the developmental processes leading to abnormal development. Therefore, no true power analysis can be performed. Instead, we assumed an 80% probability of producing 3 or more suitable embryos per treated dam as equivalent to a required beta strength of 80%.
Comment 4:
For the segmentation process, mention any quality control measures taken to ensure accurate and consistent results.
Response 4:
We thank the reviewer for this recommendation. The segmentation process is done manually and thus depends on the practical experience of the person doing the segmentation. This person should have indepth knowledge about normal embryonic organ/ structure development to be able to distinguish between normal and abnormal development in order to produce correct results. Thus, the quality control measures are ensured by D.K. (48 years of experience in embryology for pediatric surgeons). We also added the following statement in the method section (line 87-93):
The segmentation of the embryonic organs to create 3D reconstructions was performed by M.G. (9 years of experience with micro-CT scans of embryos). The analysis of the findings from the micro-CT data sets of malformed embryos was performed by D.K. (42 years of experience in pediatric surgery and 48 years in embryology for pediatric surgeons) and M.G. Cardiac malformations were assessed by Dr. med. Werner Siekmeyer, specialist in pediatric cardiology (Clinic and Polyclinic for Cardiology).
Reviewer 3 Report
Comments and Suggestions for Authors
Manuscript titled "Micro computed tomography in the Adriamycin rat model of malformations
- Preliminarily results" is a short preliminary study describing findings on embryos affected by Adriamycin. The qualitative results indicate that several malformations in embryos from ED16 to ED21 are attributed to the VACTERL association. The manuscript is well written with good illustrations but still has some shortcomings, listed below:
The findings are illustrated by a number of well-designed 3D segmented images, and the methods are mostly clearly described, except for the number of embryos utilized in this study. Was that 10 as indicated in L.70 or 31 embryos as L.81? There is no information available about the control animals or their numbers.
There are missing scale bars on all the images.
The hypothesis regarding the course of events leading to tracheal agenesis is presented in the discussion section, which should be included as part of the Results Section.
The manuscript would be more impactful if the findings (malformations) were quantified, not only showing the Adriamycin-induced pathology but also quantifying the changes.
Author Response
We thank the reviewer for his/her review and would like to answer the questions and comments point by point.
Comment 1:
The findings are illustrated by a number of well-designed 3D segmented images, and the methods are mostly clearly described, except for the number of embryos utilized in this study. Was that 10 as indicated in L.70 or 31 embryos as L.81? There is no information available about the control animals or their numbers.
Response 1:
We thank the reviewer for pointing out this mistake. Actually, we harvested a total of 31 embryos for the shown age groups and used 10 embryos for the current preliminary study. The mistake is corrected in the revised manuscript (paragraph from line 95-98). In addition we added the following scentence (line 81-83):
Three datasets of normal untreated control animals were used from Publissio ZB MED Information Centre of Life Science at https://doi.org/10.4126/FRL01-006424446 [14].
Comment 2:
There are missing scale bars on all the images.
Response 2:
We thank the reviewer for pointing this out. Scale bars are included in the revised figures.
Comment 3:
The hypothesis regarding the course of events leading to tracheal agenesis is presented in the discussion section, which should be included as part of the Results Section.
Response 3:
We thank the reviewer for this suggestion. As the reviewer uses the term "hypothesis" him/herself, we believe that it should not be part of the results section, but rather in the dicussion section. Thus, we did not change the manuscript on this matter.
Comment 4:
The manuscript would be more impactful if the findings (malformations) were quantified, not only showing the Adriamycin-induced pathology but also quantifying the changes.
Response 4:
We thank the reviewer for this question and we will include these quantified data in future studies. As the shown preliminary data may be quite different from the final results, we will not include a quantification into the current study.